# Eight Years of African Swine Fever in the Baltic States: Epidemiological Reflections

**DOI:** 10.3390/pathogens11060711

**Published:** 2022-06-20

**Authors:** Katja Schulz, Edvīns Oļševskis, Arvo Viltrop, Marius Masiulis, Christoph Staubach, Imbi Nurmoja, Kristīne Lamberga, Mārtiņš Seržants, Alvydas Malakauskas, Franz Josef Conraths, Carola Sauter-Louis

**Affiliations:** 1Friedrich-Loeffler-Institut, Federal Research Institute for Animal Health, Institute of Epidemiology, 17493 Greifswald-Insel Riems, Germany; christoph.staubach@fli.de (C.S.); franz.conraths@fli.de (F.J.C.); carola.sauter-louis@fli.de (C.S.-L.); 2Food and Veterinary Service, 1050 Riga, Latvia; edvins.olsevskis@pvd.gov.lv (E.O.); kristine.lamberga@pvd.gov.lv (K.L.); martins.serzants@pvd.gov.lv (M.S.); 3Institute of Food Safety, Animal Health and Environment-“BIOR”, 1076 Riga, Latvia; 4Institute of Veterinary Medicine and Animal Sciences, Estonian University of Life Science, 51006 Tartu, Estonia; arvo.viltrop@emu.ee; 5State Food and Veterinary Service, LT07170 Vilnius, Lithuania; marius.masiulis@vmvt.lt (M.M.); alvydas.malakauskas@lsmuni.lt (A.M.); 6Dr. L Kriauceliunas Small Animal Clinic, Veterinary Academy, Lithuanian University of Health Sciences, LT47181 Kaunas, Lithuania; 7Estonian Veterinary and Food Laboratory (VFL), 51006 Tartu, Estonia; imbi.nurmoja@vetlab.ee; 8Faculty of Veterinary Medicine, Latvia University of Life Sciences and Technologies, 3001 Jelgava, Latvia; 9Department of Veterinary Pathobiology, Veterinary Academy, Lithuanian University of Health Sciences, LT44307 Kaunas, Lithuania

**Keywords:** surveillance, epidemic curve, epidemiology, wild boar, disease control

## Abstract

African swine fever (ASF) was first detected in Lithuania, Latvia, and Estonia in 2014 and has since been circulating in the Baltic States with a similar epidemiological course characterized by persistence of the disease in the wild boar population and occasional spill-over infections in domestic pigs. The aim of the present study was to evaluate surveillance data on ASF in wild boar from the three countries to improve our understanding of the course of the disease. ASF surveillance and wild boar population data of the countries were analyzed. In all three countries, a decrease in the prevalence of ASF virus-positive wild boar was observed over time. Although somewhat delayed, an increase in the seroprevalence was seen. At the same time, the wild boar population density decreased significantly. Towards the end of the study period, the wild boar population recovered, and the prevalence of ASF virus-positive wild boar increased again, whereas the seroprevalence decreased. The decreasing virus prevalence has obviously led to virus circulation at a very low level. Together with the decreasing wild boar population density, the detection of ASF-infected wild boar and thus ASF control has become increasingly difficult. The course of ASF and its continuous spread clearly demonstrate the necessity to scrutinize current ASF surveillance and control strategies fundamentally and to consider new transdisciplinary approaches.

## 1. Introduction

In 2014, African swine fever (ASF) emerged for the first time in the three Baltic states. In Lithuania, the disease was detected in January 2014. An infected wild boar was found in eastern Lithuania, close to the border of Belarus [1]. In Latvia, where the disease was detected in June 2014, and in Estonia, where first infected wild boar were reported in September 2014, the sites where the first cases were detected were located close to the border of affected neighboring countries, i.e., Belarus and Latvia [2,3]. It was therefore hypothesized that migrating wild boar infected with ASF virus (ASFV) had introduced the disease into the Baltic states [2,3]. In all three countries, the disease spread continuously, affecting almost the whole territories of Latvia and Estonia after two to three years and Lithuania after six years.

Although wild boar cases have clearly dominated the epidemic so far, numerous domestic pig outbreaks occurred in all three countries [4]. However, over three years (2018–2020), Estonia reported no ASF outbreaks in domestic pigs, whereas in Latvia, domestic pig outbreaks occurred in all years since the country became affected. In Lithuania, the highest number of ASF outbreaks in domestic pigs was reported in 2018 (51 outbreak), when the highest number of wild boars infected with ASF was found, and since then, a decrease in the cases as well in the outbreaks was observed, with nineteen outbreaks in 2019 and three outbreaks in 2020. In 2021, no such outbreaks occurred in the country [5].

The role of wild boar in the current ASF epidemic, the continuous threat that infected wild boar populations pose to the global pig industry, and the resulting requirement of effective surveillance and control measures are undisputed [6]. However, despite strong efforts and intensive research, it was so far not possible to eliminate the disease from the Baltic states. The epidemiological course in the three countries has so far been similar [7,8,9]. However, in Estonia, there was hope for more than a year that the country might have managed to eliminate the disease [10]. Nevertheless, new ASFV-positive wild boar emerged in summer 2020. The reasons for the re-occurrence of the disease could not be clearly identified [11]. At the latest, when ASF broke out in an Estonian domestic pig holding in summer 2021, it became clear that ASF was still present in the country.

In all three countries, participatory methods were used to seek contact to hunter communities and to evaluate their perceptions regarding the surveillance and control of ASF in wild boar [12,13,14]. With the help of these studies, barriers and starting points to increase the willingness of hunters to support ASF surveillance and control could be identified. These insights may support the design of surveillance and control measures that might be received more favorably by hunters and supported. This may increase the chances for successful ASF control in wild boar populations.

In continuation of previous studies, the aim of this study was to analyze the course of ASF in Estonia, Latvia, and Lithuania by integrating recent surveillance and wild boar population data and to describe possible scenarios of the further course of ASF. We expected to identify similar patterns and potential weaknesses in surveillance and control to improve measures and thus to increase the chances of ASF control in wild boar. The study results can not only support disease control in the Baltic countries but may also help other affected countries in adapting control measures based on the experiences from Estonia, Latvia, and Lithuania.

## 2. Results

### 2.1. Descriptive Data

For Estonia, 62,944 data records over 84 study months were available, of which 60,238 originated from active surveillance (from hunted wild boar) and 2706 from passive surveillance (from wild boar found dead, shot sick, or involved in a road traffic accident (RTA)). For Latvia, 102,321 data records over 83 study months were analyzed. Of the analyzed samples, 99,665 came from active and 2656 from passive surveillance. Lithuanian data were available for 72 study months. In total, 87,307 data records were analyzed, of which 83,566 data came from active and 3741 from passive surveillance.

In Estonia and Lithuania, the highest number of samples from active surveillance was investigated in 2016, and after a lower number of samples in the following years, the number of samples increased again in 2021. In Latvia, the number of samples was very similar in all years, but in 2020 and 2021, the number of samples originating from active surveillance increased (Figure 1). In all three countries, the number of samples from passive surveillance was highest in the first three years of the study period (Figure 2). Accordingly, the number of samples (from active and passive surveillance) increased again in 2021 in the three Baltic states. Furthermore, the seasonal pattern of sample size was similar in all three countries. In each year, the highest number of samples originating from hunted wild boar were investigated in January, and the lowest number of samples was available in April (Figure 1). Samples from passive surveillance were mainly taken in July (Latvia and Lithuania) and in December (Estonia). The fewest samples were taken in June in all three countries (Figure 2).

### 2.2. Model Analysis

Model analyses revealed a similar temporal course of the ASF prevalence in the three countries. The ASFV prevalence in hunted wild boar and in wild boar found dead increased in the beginning of the study period and started to decrease in May 2017 (Estonia), in the middle of 2018 (Latvia) and in May 2019 (Lithuania). However, the ASFV prevalence in hunted wild boar and wild boar found dead slightly increased again in all three countries in the last year of the study period (Figure 3 and Appendix A). In all Baltic states, the seroprevalence increased somewhat delayed with respect to the increase of the prevalence of ASFV-positive wild boar. Moreover, the subsequent decrease was slightly delayed (Figure 4).

### 2.3. Prevalence Estimates for Different Age Classes

In hunted wild boar, the median ASFV prevalence was highest in animals younger than 1 year, and the median seroprevalence reached its maximum in animals older than 2 years. In contrast, the median ASFV prevalence in wild boar found dead showed the highest value for animals between 1 and 2 years. However, none of these differences was statistically significant (Table 1, Appendix A).

### 2.4. Wild Boar Population Density

In Estonia, the median estimated number of wild boar/km^2^ was similar from hunting seasons 2012/13 to 2015/16. Already in 2016/17, a decrease, albeit not statistically significant, was observed. The population density decreased significantly from hunting season 2015/16 to 2017/18 and in the following years. From 2018/19 to 2019/20 and from 2019/20 to 2020/21, the number of wild boar/km^2^ increased statistically significantly (Table 2 and Appendix A).

Regarding the geographical distribution of the wild boar population density in Estonia, the population density seemed to increase at the beginning of the epidemic in the hunting season 2014/15 and 2015/16. In the following years, the population density first dropped in the affected areas in eastern and southern Estonia. In 2018/19, it was extremely low in almost the whole country except for the two islands in the west. In almost all Estonian counties but particularly in the southeast, the wild boar population density increased in the hunting season 2020/21 (Appendix A).

In Latvia, the temporal course in the population density was similar to the one in Estonia. However, the drop in the wild boar population density was already statistically significant from the hunting season 2015/16 to 2016/17 onwards. In hunting seasons 2019/20 and 2020/21, the wild boar population density started to increase (Table 3 and Appendix A).

In Latvia, the wild boar population densities in the individual hunting management units increased, like in Estonia, at the beginning of the epidemic. Additionally, comparable to Estonia, it decreased over time depending on the ASF status of the area. In the hunting season 2019/20, the wild boar population density slowly increased again, particularly in the eastern and central part of the country. This course continued, and in hunting season 2020/21, the population density further increased, particularly in the center and in the eastern part of the country (Appendix A).

## 3. Discussion

For this study, we used comprehensive real-life wild boar surveillance data to evaluate the epidemiological course of ASF in the three Baltic countries from the beginning of the epidemic until the end of the year 2021, i.e., over approximately eight years, thus expanding previous studies [7,8,9,10,15]. The current course of ASF and its further spread within the Baltic countries and beyond make it necessary to study the epidemiology of the disease over time and to learn from the experience made with particular control measures and their implementation. Continuous scientific analysis that includes the most recent data can help to scrutinize existing hypotheses on the course of ASF in wild boar to improve surveillance and control measures and thus to increase the chance for successful and sustainable control. To warrant comparability with previous studies, we largely used established methods [7,8,9,10,15].

Similar to other countries [16], the number of samples originating from hunted wild boar was clearly higher than the number of those from passive surveillance in all three countries during the entire study period. Detecting wild boar that died from ASF is not easy, as sick and dying animals normally seem to retreat as far as possible or may be carried off or eaten by other wild animals [17,18]. However, Gervasi and Gubertì [19] showed in their model analyses that it would require at least 5–15 carcasses for each 100 hunted wild boar to be removed to have a chance to successfully break the transmission cycle of ASF and thus to potentially eliminate ASF from an affected area. In restricted areas [20], the sampling of all found wild boar carcasses is mandatory. Thus, in all three countries, the low number of yearly samples originating from passive surveillance already indicate the poor chances to successfully eliminate ASF from the wild boar population but could also indicate the low ASFV circulation in the infected areas. In addition, landscapes such as huge forests and wetlands can impede carcass search and allow the virus to spread undisturbed.

Lithuanian surveillance data were only used from 2016 onwards, which was due to the unavailable geographical data in the datasets recorded before. The larger national territory of Lithuania and Latvia, some unaffected areas, and the bigger wild boar population are probably factors leading to bigger samples sizes from the two countries compared to Estonia. Additionally, expansion of the epidemics has been different in the Baltic countries. The sampling intensity of hunted wild boar changed several times depending on the ASF status of the affected area. These changes were due to updates of European Commission Implementing Decision 2014/709/EU (replaced by the Commission Implementing Regulation (EU) 2021/605 as of 21 April 2021 [20]) and regionalization measures applied in the countries. Furthermore, hunting networks, personal hunting habits, and also environmental factors may have influenced the numbers of samples investigated in the countries.

The higher number of samples from passive surveillance in Lithuania was mainly prominent from 2016 to 2018. This could be due to the incentives that were paid to the general public for the reporting of wild boar carcasses. In Latvia, such incentives were also paid but only from 2014 to 2015. In June 2020, the Latvian authorities started to pay those incentives again, thus positively influencing the number of samples. The variations in the number of samples originating from passive surveillance certainly partly reflect the impacts of different motivational measures.

The larger number of investigated samples at the beginning of the epidemic was probably due to several factors. It can be assumed that the motivation and the hope to eliminate the disease successfully resulted in an increased effort to hunt or find and sample wild boar. Moreover, the population density was significantly higher shortly after the introduction of ASF than in subsequent years so that more wild boar were available for sampling. The decline in the number of samples obtained from dead wild boar has been described to indicate a late phase of the epidemic [10]. It is known, however, that hunters do not fully support passive surveillance [12,13,14,21,22], still emphasizing the potential role of increasing fatigue and demotivation among hunters who were expected to support passive surveillance activities for a long time.

By contrast, the increasing number of samples in the last study year is mainly due to a recovery of the wild boar population. Furthermore, it cannot be ruled out that the motivation of the involved stakeholders has increased recently since they may have gotten the impression that they were close to the goal of eliminating ASF. In addition, in Latvia, the reintroduction of incentives for informing the authorities about the detection of dead wild boar since July 2020 has probably heightened the motivation of hunters or the general public to support passive surveillance.

The seasonal patterns observed in the sample size are plausible for the following reasons. The small number of samples obtained from hunted wild boar in April each year can be attributed to the end of the active hunting season (driven hunts usually take place between October and February) and the main reproduction month of wild boar. In contrast, the large number of samples originating from hunted animals in January each year can be explained by the increased hunting activity [23,24]. January and February have historically and traditionally been the main hunting period. The large number of samples from passive surveillance in July, preceded by a low number of samples in June, is not easy to explain. This is particularly interesting since wild boar carcasses decompose much faster during summer months, suggesting a lower detection probability of wild boar carcasses [25].

However, the larger numbers of samples might be due to the high case/fatality ratio of ASF in young wild boar. In addition, this “new generation” increases the population density during these months, and the piglets are already old enough to move around actively and also big enough to be detected in case of their death. The higher number of samples may also be related to the summer holidays and thus to increased leisure activities in the forests.

The temporal course of the prevalence of ASFV-positive wild boar and of the seroprevalence was similar in the three countries. It was characterized by an increasing ASFV prevalence at the beginning of the epidemic and a decrease in the subsequent 3–4 years. Interestingly, the ASFV prevalence increased again at the end of the study period emphasizing the increasing circulation of ASFV within the wild boar population. The increasing seroprevalence may indicate an accumulation of wild boar that survived the disease during the course of the epidemic. The decrease in the seroprevalence that followed the decreasing prevalence of ASFV-positive wild boar seemed to be just a logical consequence of the smaller number of ASFV-positive wild boar, which led to a smaller number of seroconversions and thus in a lower seroprevalence.

In a Bayesian model, we included prevalence estimates for neighboring administrative units and earlier time points. This approach makes the results of the model analysis more robust and less prone to bias. When comparing the whole period of the epidemic, the epidemiological waves in Latvia were not as pronounced as in Estonia and in Lithuania, which could indicate a slower spread of ASF in Latvia. However, recent analyses showed a slower ASF spread in Lithuania, but only data until summer 2016 were analyzed [26].

Although age-dependent variations of clinical signs and severity of disease are reported [27,28], no significant differences in the prevalence of ASFV-positive wild boar and the seroprevalence in different age classes were found. These results emphasize the need to hunt and sample all wild boar regardless of their age.

Due to the lack of consistent methods to determine wild boar population densities [6], available population density data generated through different methods such as hunting bag statistics, snow-track counts, sightings, and hunter estimations were used for the analyses. Although these data may not be as accurate as desired and make a direct comparison between countries difficult, they are the only available data. Our comparison revealed the same trend in the dynamics of the wild boar population density in Estonia and Latvia and also in Lithuania [9].

In Estonia, Latvia, and Lithuania [9], the wild boar population density decreased significantly after a few years of ASFV circulation. Morelle et al. [29] showed that this reduction is probably caused through ASF itself rather than by increased hunting as a measure to control ASF in the wild boar population. These findings are supported by the almost absent change in the population density on the Estonian island of Hiiumaa over the years. The island has never been affected by ASF, and its wild boar population density remained stable despite of increased hunting efforts.

The geographical analyses of the hunting bag data showed an increase of the wild boar density at the beginning of the epidemic, indicating an increased hunting effort. The increase of the population density by the end of the study period could mainly be observed in areas, where ASF was present at the beginning of the epidemic, thus suggesting a recovery of the wild boar population when the number of new ASFV infections in wild boar had decreased. Therefore, the number of susceptible hosts grow again, thus keeping the infection cycle alive.

The re-emergence of ASFV-positive wild boar in Estonia after a long period without any reported ASFV-positive wild boar clearly illustrates the huge challenges for a sustainable ASF control. Different hypotheses for potential causes of this re-occurrence were tested, but unfortunately, no clear answer could so far be provided [11].

The epidemiological analyses of more than 200,000 wild boar samples help to identify one of the main challenges in the control of ASF in wild boar. The analyses of real-life data offer the unique opportunity to provide concrete evidence and thus to purposeful determine starting points for a more effective disease control. After several years of the epidemic, the wild boar population density dropped significantly. Thus, the number of susceptible hosts decreased, and so did the prevalence of ASFV-positive wild boar. This course makes the detection of circulating virus almost impossible [11,30]. As a consequence of the low prevalence of ASFV-positive wild boar, the wild boar population recovered, thus leading to a larger number of more susceptible individuals, new infections, and further virus spread. This epidemiological course can be described using a conventional susceptible-infected-recovered (SIR) model and was observed in other wildlife infectious diseases [31]. However, in contrast to classical swine fever, which is highly infectious [32], the spread of ASF is rather slow [33]. Thus, to break the described vicious circle, more knowledge is required about the mechanisms that support virus persistence at a low level despite a small number of susceptible hosts. Further research is necessary to clarify the role of contacts between the individual wild boar packs, the role of carcasses, and their potentially infectious direct environment [34,35,36]. Although O’Neill et al. [37] used a modelling approach, they also identified the urgent need to further investigate the role of carcasses, seropositive wild boar, and the population density. The effects of different control strategies should be further evaluated [19] and the results scientifically but also politically communicated. In addition, it is inevitable to motivate all stakeholders involved in ASF control for a long-term commitment. Awareness must not only be raised but also maintained and resources allocated to cope with the described challenges and the huge transdisciplinary effort that is needed. Stakeholders and decision makers should be made aware that the epidemic will probably not be eliminated within a short time. This applies particularly to countries that face a situation similar to the one in the Baltic states, where continuous infection pressure from neighboring countries increases the risk of new virus introductions [38].

A potentially successful ASF control should be based on several pillars. No stone should be left unturned in reducing the number of wild boar both in affected areas and in those that are still free from ASF. Thus, further research is necessary identifying effective and sustainable methods to reduce the wild boar population density. Further epidemiological research is necessary to understand the drivers of the disease. This research should be based on interdisciplinarity including wildlife biologists, epidemiologists, and virologists. In addition, transdisciplinary communication and cooperation is essential. Scientific evidence should be communicated to decision makers and perceptions of key figures considered. Following this evidence, decision makers should be courageous and, if necessary, leave familiar paths to try out new approaches. However, in situations, where ASF-infected wild boar can cross national borders at any time, the long-term control of ASF in wild boar is very likely to stay a challenge.

## 4. Materials and Methods

### 4.1. Descriptive Data Analysis

Data of all three countries originated from the CSF/ASF wild boar surveillance database of the European Union (https://surv-wildboar.eu, accessed on 1 January 2022). The data were used with the approval of the relevant authorities. All data records included information about the age of the sampled animal (<1 year, 1–2 years, >2 years), the place and time of sampling, and the origin of the sample (active surveillance: hunted wild boar; passive surveillance). The number of samples from passive surveillance was composed of samples from wild boar found dead, shot sick, and killed in RTA. However, due to the small number of wild boar samples originating from animals shot sick (Estonia: 27, Latvia: 0, Lithuania 0) and from wild boar killed in an RTAs (Estonia: 189, Latvia: 21, Lithuania 31), all data were categorized as passive surveillance. Furthermore, the laboratory test results were available from the database. Detailed information about sampling and laboratory testing have been published elsewhere [1,7,8].

For Estonia and Latvia, analyzable data were available from 1 January 2015, whereas for Lithuania, data were used from 1 January 2016. For Estonia and Lithuania, data were available until the 30 December 2021 and for Latvia until the 30 November 2021.

For each month of the year, the numbers of samples originating from active and passive surveillance were determined. All data records that lacked information on the origin of the sample (active or passive surveillance) were excluded from the analysis. Figures were generated using the package “ggplot2” and “cowplot” of the software R (https://www.r-project.org/, accessed on 3 May 2022).

### 4.2. Model Analysis

The temporal course of the prevalence estimates was calculated using a Bayesian space–time model and BayesX 2.0.1 (http://www.uni-goettingen.de/de/bayesx/550513.html, accessed on 3 May 2022). The calculations were done for seroprevalence estimates (number of wild boar that had tested positive for ASFV-specific antibodies and were at the same time negative for ASFV divided by the total number of wild boar investigated serologically) and for ASFV prevalence estimates (number of wild boar that had tested ASFV-positive irrespective of the serological test result divided by the total number of wild boar investigated for ASFV). Seroprevalence calculations were only done for hunted wild boar, and ASFV prevalence calculations were stratified for hunted wild boar and wild boar found dead.

The temporal course of the three different prevalence categories was determined on a monthly basis, essentially as described in previous studies [3,8,10]. Within the model, age was categorized as two classes (<2 years and >2 years) and constituted the fixed independent variable. The prevalence of ASFV-positive wild boar or the seroprevalence estimates, respectively, represented the dependent variable. Time, space, and season were included as random factors (Appendix A). The model is described in more detail elsewhere [8,39,40]. Figures were generated using R (https://www.r-project.org/, accessed on 3 May 2022).

### 4.3. Prevalence Estimates for Different Age Classes

Investigating the differences in the seroprevalence and in the prevalence estimates of ASFV-positive wild boar between different age classes (<1 year, 1–2 years, >2 years), surveillance data of the three countries were merged. Again, seroprevalence estimates were only calculated for hunted wild boar, and ASFV prevalence estimates were calculated for hunted wild boar and for wild boar found dead. Differences in the yearly prevalence estimates between the different age classes were calculated and analyzed using the non-parametric Kruskal–Wallis test. A *p*-value < 0.05 was considered statistically significant. Prevalence estimates and statistical analyses were performed also using R (https://www.r-project.org/, accessed on 3 May 2022).

### 4.4. Wild Boar Population Density

Integrating most recent data, the changes of the wild boar population density (number of wild boar/km^2^) over time were analyzed for Estonia and Latvia. In line with recent analyses of the Estonian wild bar population [8,10], data were analyzed on county level and from hunting season 2012/13–2020/21. For Latvia, data on hunting management unit and from hunting season 2014/15–2020/21 were analyzed as described [41]. The differences in the wild boar population density were determined using the Kruskal–Wallis test. For pairwise comparisons, the Mann–Whitney U test was used, and the Bonferroni correction applied to control for type I-error [42]. A *p*-value < 0.05 was considered statistically significant.

## Figures and Tables

**Figure 1 pathogens-11-00711-f001:**
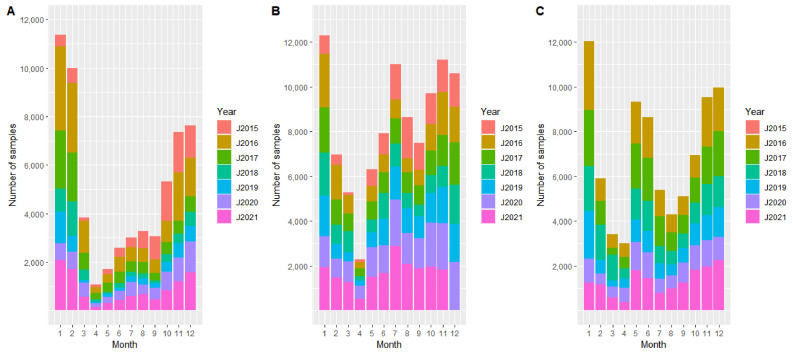
Number of investigated samples from hunted wild boar (active surveillance) from Estonia (**A**), Latvia (**B**), and Lithuania (**C**) per months of the years 2015–2021. For Lithuania, no data were available for 2015. For Latvia, no data from December 2021 were available.

**Figure 2 pathogens-11-00711-f002:**
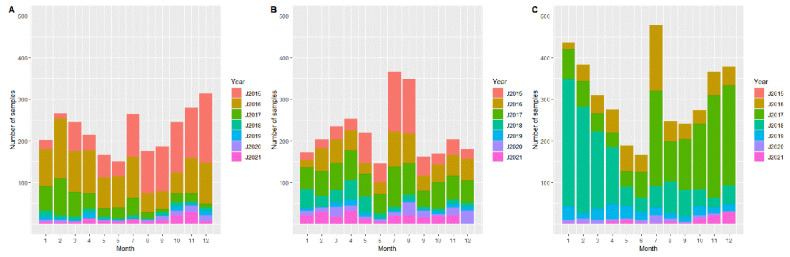
Number of investigated samples from wild boar found dead (passive surveillance) from Estonia (**A**), Latvia (**B**), and Lithuania (**C**) per months of the years 2015–2021. For Lithuania, no data were available for 2015. For Latvia, no data from December 2021 were available.

**Figure 3 pathogens-11-00711-f003:**
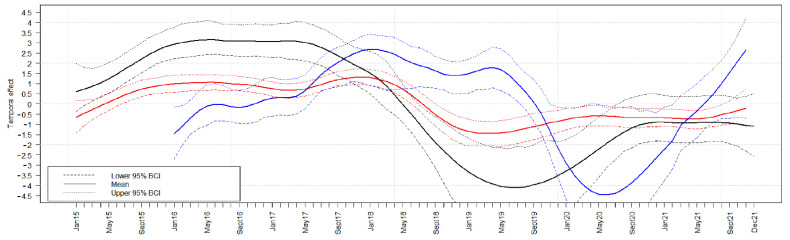
Median temporal effect on the logit prevalence for wild boar found dead that tested positive for ASFV in Estonia (black lines), Latvia (red lines), and Lithuania (blue lines). The 95% Bayesian credible intervals (BCIs, dashed lines) are indicated for each country in the respective color.

**Figure 4 pathogens-11-00711-f004:**
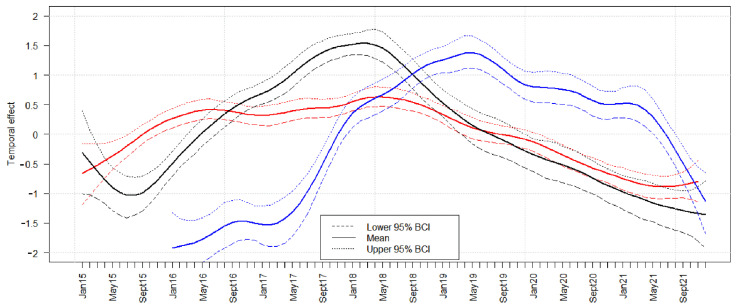
Median temporal effect on the logit seroprevalence for hunted wild boar in Estonia (black lines), Latvia (red lines), and Lithuania (blue lines). The 95% Bayesian credible intervals (BCIs, dashed lines) are indicated for each country in the respective color.

**Table 1 pathogens-11-00711-t001:** Median ASFV and seroprevalence estimates in hunted wild boar and ASFV prevalence estimates in wild boar found dead in the different age classes, including the *p*-values calculated using the Kruskal–Wallis test.

	Age Class			
Median in % of	<1 Year	1–2 Years	>2 Years	*p*-Value
Seroprevalence	1.44	1.57	1.60	0.59
Virus prevalence in hunted wild boar	2.95	1.66	0.87	0.16
Virus prevalence in wild boar found dead	50.41	55.29	45.67	0.89

**Table 2 pathogens-11-00711-t002:** Median of the estimated number of Estonian wild boar/km^2^ per hunting season and statistical analyses of differences between hunting seasons (years).

Hunting Season	Median of the Estimated Number of Wild Boar/km^2^	Hunting Season							
		2013/14	2014/15	2015/16	2016/17	2017/18	2018/19	2019/20	2020/21
		*p*-Value *							
2012/13	0.58	1	1	1	1	0.03	<0.001	<0.001	0.05
2013/14	0.54		1	1	1	0.06	<0.001	<0.001	0.04
2014/15	0.60			1	0.85	0.03	<0.001	<0.001	0.04
2015/16	0.67				0.52	0.005	<0.001	<0.001	0.003
2016/17	0.19					1	1	0.60	1
2017/18	0.09						1	1	0.18
2018/19	0.08							1	0.006
2019/20	0.07								0.01
2020/21	0.17								

* calculated using the Mann–Whitney U test with a correction for multiple pairwise testing.

**Table 3 pathogens-11-00711-t003:** Median of the estimated number of Latvian wild boar/km^2^ per hunting season and statistical analyses of differences between hunting seasons (years).

Hunting Season	Median of Estimated Number of Wild Boar/km^2^	Hunting Season					
		2015/16	2016/17	2017/18	2018/19	2019/20	2020/21
		*p*-Value *					
2014/15	0.66	1	<0.001	<0.001	<0.001	<0.001	<0.001
2015/16	0.65		<0.001	<0.001	<0.001	<0.001	
2016/17	0.25			<0.001	<0.001	<0.001	1
2017/18	0.16				<0.001	1	<0.001
2018/19	0.11					<0.001	<0.001
2019/20	0.17						<0.001
2020/21	0.26						

* calculated using the Mann–Whitney U test with a correction for multiple pairwise testing.

## Data Availability

The original data used for the analyses can be obtained from the corresponding author after approval by the responsible institutions in Estonia, Latvia, and Lithuania.

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
