# Peer review of "Eight Years of African Swine Fever in the Baltic States: Epidemiological Reflections"

_pathogens, 2022, doi:10.3390/pathogens11060711_

Round 1
Reviewer 1 Report
This study, based on the real-life wild boar surveillance data, comprehensively evaluated the epidemiological course of ASF in three Baltic countries (Lithuania, Latvia and Estonia) for a period over approximately 8 years. As described in the manuscript, the molecular prevalence and seroprevalence of ASFV in wild boar from the three countries were varied over time, which were affected by the wild boar population density (the number of samples), seasonal patterns and age effects. In general, the manuscript is well written, and improves the understanding of the course of ASF in the three Baltic countries, except for some concerns as outlined for the authors to consider below.
1, lines 33-35, the meaning of the sentence “the declines in virus prevalence and wild boar population density makes the detection of ASF-infected wild boar and thus ASF control increasingly difficult” is not clear or accurate.
2, the authors observed a decrease in the prevalence of ASFV-positive wild boar as long as the wild boar population decreased, and an increase in the prevalence of ASFV-positive wild boar as long as the wild boar population recovered. Were the calculations conducted only based on the numbers of ASFV-positive wild boar or the ASFV-positive rates (the number of ASFV-positive wild boar/total wild boar)? Please clarify it in the manuscript.
Author Response
This study, based on the real-life wild boar surveillance data, comprehensively evaluated the epidemiological course of ASF in three Baltic countries (Lithuania, Latvia and Estonia) for a period over approximately 8 years. As described in the manuscript, the molecular prevalence and seroprevalence of ASFV in wild boar from the three countries were varied over time, which were affected by the wild boar population density (the number of samples), seasonal patterns and age effects. In general, the manuscript is well written, and improves the understanding of the course of ASF in the three Baltic countries, except for some concerns as outlined for the authors to consider below.
Thank you for your positive feedback
1, lines 33-35, the meaning of the sentence “the declines in virus prevalence and wild boar population density makes the detection of ASF-infected wild boar and thus ASF control increasingly difficult” is not clear or accurate.
We changed the text for a better understanding.
Lines 33-36: The decreasing virus prevalence has obviously led to virus circulation at a very low level. Together with the decreasing wild boar population density, the detection of ASF-infected wild boar and thus ASF control has become increasingly difficult.
2, the authors observed a decrease in the prevalence of ASFV-positive wild boar as long as the wild boar population decreased, and an increase in the prevalence of ASFV-positive wild boar as long as the wild boar population recovered. Were the calculations conducted only based on the numbers of ASFV-positive wild boar or the ASFV-positive rates (the number of ASFV-positive wild boar/total wild boar)? Please clarify it in the manuscript.
We clarified the calculations of prevalence estimates in the Material and Methods section.
Lines 364-369: The calculations were done for seroprevalence estimates (number of wild boar that had tested positive for ASFV-specific antibodies and were at the same time negative for ASFV divided by the total number of wild boar investigated serologically), and for ASFV prevalence estimates (number of wild boar that had tested ASFV-positive, irrespective of the serological test result, divided by the total number of wild boar investigated for ASFV).

Reviewer 2 Report
ASF is an economically important infectious disease for global pig industry. Up to date, the commercial vaccine is still unavailable, and any treatment is forbidden, therefore the only way to control the disease is provided by strict sanitary measures and passive and active surveillance on wild boar. The aim of the present study was to evaluate surveillance data on ASF in wild boar from the three countries: Lithuania, Latvia and Estonia to improve our understanding of the course of the disease. In the reviewed article, a very detailed statistical analysis of the available data was made, the conclusions drawn regarding the surveillance of ASF and the population of wild boar in ASF-affected countries are very valuable and can serve to improve the methods of preventing the spread of this disease.
Line 65, 118,250: Maybe “however” than “yet” will be proper word
Line 87: “active surveillance” should be short explained e.g. (hunted wild boar) and the same situation with “passive surveillance”.
Line 460: one dot too many.
Author Response
ASF is an economically important infectious disease for global pig industry. Up to date, the commercial vaccine is still unavailable, and any treatment is forbidden, therefore the only way to control the disease is provided by strict sanitary measures and passive and active surveillance on wild boar. The aim of the present study was to evaluate surveillance data on ASF in wild boar from the three countries: Lithuania, Latvia and Estonia to improve our understanding of the course of the disease. In the reviewed article, a very detailed statistical analysis of the available data was made, the conclusions drawn regarding the surveillance of ASF and the population of wild boar in ASF-affected countries are very valuable and can serve to improve the methods of preventing the spread of this disease.
Thank you very much for the positive evaluation of our manuscript.
Line 65, 118,250: Maybe “however” than “yet” will be proper word
In line 66, we changed the word to “nevertheless” as “however” was already used in the preceding sentence. In lines 119 and 251, we changed “yet” to “however”.
Line 87: “active surveillance” should be short explained e.g. (hunted wild boar) and the same situation with “passive surveillance”.
The description is already included in the Material and Methods section (line 346-348). In the readers’ interest, we included the explanation also in the Results section (line 88-90).
Line 460: one dot too many.
Thank you, we deleted the dot.

Reviewer 3 Report
I reviewed the manuscript “Eight years of African Swine Fever in the Baltic States: Epidemiological reflections”. In this epidemiological study authors describe the prevalence of ASF in wild boar in Lithuania, Latvia, and Estonia.
Overall, I found this manuscript interesting considering all the uncertainties associated with the control of ASFV. I consider that the results all were supported considering the high number of cases used in this study.
These are some of my suggestions to improve the quality of this study.
-based on the available data in the three countries, is it possible to establish a statistical correlation between the results presented herein in wild boar and the number of outbreaks in domestics in each country?
-If possible, I suggest adding a final figure representing and developing more the concept presented in the statement between lines 329-331. In the text develop more the concept in terms of epidemiology.
-Increase the discussion section by contrasting the results with previous publications in other countries. Discuss about the methodology used to develop this work versus the methodology presented in other studies like “Modelling the transmission and persistence of African swine fever in wild boar in contrasting European scenarios” O’Neill et al., 2020.
Author Response
I reviewed the manuscript “Eight years of African Swine Fever in the Baltic States: Epidemiological reflections”. In this epidemiological study authors describe the prevalence of ASF in wild boar in Lithuania, Latvia, and Estonia.
Overall, I found this manuscript interesting considering all the uncertainties associated with the control of ASFV. I consider that the results all were supported considering the high number of cases used in this study.
Thank you for your positive feedback.
These are some of my suggestions to improve the quality of this study.
-based on the available data in the three countries, is it possible to establish a statistical correlation between the results presented herein in wild boar and the number of outbreaks in domestics in each country?
In the present manuscript, we exclusively investigated the epidemiological course of ASF in wild boar, without considering any outbreaks in domestic pig holdings. Any potential correlations between outbreaks in domestic pigs and the cases in wild boar would require close scrutiny and a separate study. We feel that this is beyond the scope of the present manuscript, which we would like to keep focused to the ASF situation in wild boar. In addition, in the study of Nurmoja et al (Nurmoja I, Mõtus K, Kristian M, Niine T, Schulz K, Depner K, Viltrop A. Epidemiological analysis of the 2015-2017 African swine fever outbreaks in Estonia. Prev Vet Med. 2020 Aug;181:104556. doi: 10.1016/j.prevetmed.2018.10.001. Epub 2018 Oct 9. PMID: 30482617), we partly addressed this question already.
-If possible, I suggest adding a final figure representing and developing more the concept presented in the statement between lines 329-331. In the text develop more the concept in terms of epidemiology.
We added more specific suggestions dealing with the epidemiological concept, but we would like to refrain from adding another figure illustrating this theoretical concept.
Lines 332-337: Thus, further research is necessary identifying effective and sustainable methods to reduce the wild boar population density.
Further epidemiological knowledge gaps are already mentioned in lines 318-322. Also, the epidemiological concept of the course of ASF in wild boar is intensively discussed in lines 304-329.
-Increase the discussion section by contrasting the results with previous publications in other countries. Discuss about the methodology used to develop this work versus the methodology presented in other studies like “Modelling the transmission and persistence of African swine fever in wild boar in contrasting European scenarios” O’Neill et al., 2020.
We added more information.
Lines 320-322: Although O’Neill, et al [37] used a modelling approach, they also identified the urgent need to further investigate the role of carcasses, seropositive wild boar and the population density.

Round 2
Reviewer 3 Report
Thanks to the authors for their responses, at this point I don't have more concerns about this study.